# Modelling and Experimental Investigation on the Settling Rate of Kaolinite Particles in Non-Ideal Sedimentation Stage under Constant Gravity

**DOI:** 10.3390/ma13173785

**Published:** 2020-08-27

**Authors:** Jianfu Wang, Xin Kang, Chunyin Peng

**Affiliations:** 1College of Civil Engineering and Architecture, Zhejiang University, Hangzhou 310058, China; 21712010@zju.edu.cn; 2College of Civil Engineering, Hunan University, Changsha 410012, China; 3College of Civil Engineering, China Three Gorges University, Yichang 443002, China

**Keywords:** non-ideal sedimentation, non-ideal sedimentation of clay particles in water system under constant gravity, inhomogeneity, instantaneous sedimentation rate, sedimentary characteristics

## Abstract

We compared the catalytic effects of two polymers (soluble starch and apple pectin) on the flocculation of kaolinite suspension. Moreover, the relationship between the zeta potential value and the time when kaolin particle sedimentation occurred was verified, and the mechanism of flocculation was analyzed. Additionally, a constitutive model was proposed to simulate the non-ideal sedimentation of clay particles in an aqueous system under constant gravity. This model not only considers the inhomogeneity of the solute but also simulates the change in clay concentration with time during the deposition process. This model proposes a decay constant (α) and sedimentation coefficient (*s*). The model can also be used to calculate the instantaneous sedimentation rate of the clay suspensions at any time and any depth for the settling cylinder. These sedimentary characteristics were simulated by adopting the established deposition model. The results show that the model is capable of predicting the time required for the complete sedimentation of particles in the aqueous system, suggesting the feasibility of engineering wastewater treatment, site dredging, etc.

## 1. Introduction

The sedimentation of silt clay, the consolidation of dredged soil, and the density and deformation control during the construction of earth-rock dams and cofferdams are closely related to the flocculation and settlement of clay. In nature, clay particles usually carry negative charges on their surface. When metal cations are dissolved in water, they neutralize the negative charges on the surface of clay particles and change the interactions between clay particles, such as Coulomb repulsion, double-layer repulsion, and Johannes D. van der Waals attraction. Adachi et al. [1] applied the scheme of DLOV theory (named after Derjaguin, Landau, Verwey, and Overbeek and is the quantitative explanation of charged colloid stability) and the concept of fractal structure of flocs to the suspension of montmorillonite; they put forward that soil particles are always present in the electrostatically dispersed regime and the coagulated regime, and the study revealed the unique nature of this clay dispersion. Additionally, because of the nature of the ionization effect in the aftermath of organic matter in water, macromolecular organic matter, through hydrogen bonding, adsorbs to the surface of the clay particles, prompting clay particle flocculation, affecting the precipitation process of clay particles and the formation of a different microstructure [2], resulting in the physical and mechanical properties of the soil being significantly different. The sedimentation process of soil is closely related to its ultimate mechanical properties. Many scholars have studied the sedimentation properties of clay particles [3,4,5,6,7,8,9,10,11,12,13,14,15,16,17,18] and have put forward relevant formulas for calculating the sedimentation rate of tiny particles. Brostow et al. [19] proposed a flocculation model by assuming that there is a relationship between the radius of rotation and the velocity of the particles after they leave the suspension. Based on this, Brostow et al. [20] proposed a treatment of industrial wastewater by polymeric flocculants, and they discussed the flocculation efficacy of polysaccharides providing both shear stability and biodegradability. However, some of these studies assume that the precipitation of soil particles occurs in uniform solute and the soil particles are regular spheres; these assumptions are not completely in line with the reality and cannot accurately reflect the real situation of clay particle deposition. Therefore, knowing how to accurately predict the deposition rate and process of clay particles is an urgent problem that needs to be solved.

Previous studies on sedimentation properties typically use the method of measuring the boundary scale between the supernatant and kaolinite suspension in the measuring cylinder to simulate the sedimentation process of soil particles. Cao et al. [21] took the zeta potential as the standard to measure the degree of mutual attraction between particles, and they studied the effect of PAM (polyacrylamide), PAA (polyacrylic acid), chitosan, and xanthan gum on the flocculation of kaolinite particles. Kang et al. [22] proposed the S-model and put forward the calculation method of the kaolinite particles’ sedimentation velocity. Here, the measuring cylinder in the kaolinite particles’ deposition changes according to the spatial distribution and, in turn, is divided into the supernatant fluid area suspension and sediment consolidation, which define the supernatant fluid area and suspension area boundary between the ratio of the height and time or the settling curve, and simulate the sedimentary characteristics under constant gravity. Kang et al. [23] used a self-developed floating wall type consolidometer bender element testing system to study the precipitation and consolidation properties of the modified kaolinite. Michaels et al. [24] investigated the settling rates and sediment volumes of kaolinite suspension; they assumed that in a flocculated suspension, the basic flow units are small dusters of particles called floes. There are two key problems in the above studies. First, the studies assume that soil particles are uniformly distributed in the suspension interval below the boundary. The assumption is inconsistent with the actual situation. Second, because of the intermolecular interaction between small particles, there is no obvious difference in the soil particle concentration distribution in the measuring cylinder, so the boundary between the supernatant and suspension cannot be observed clearly. Therefore, it is difficult to accurately describe the sedimentation process of clay particles in the measuring cylinder.

The sedimentation and diffusion process of small particles can be analyzed by the Lamm equation. The non-ideal sedimentation model based on the Lamm equation can effectively solve the problems of the uneven distribution of small particles and address the difficulty in the observation of the boundary between the supernatant and suspension. Solovyova et al. [25] used a non-ideal sedimentation model based on the Lamm equation to explain the weak interaction between particles, and they proposed a relationship between the sedimentation coefficient (*s*) and diffusion coefficient (*D*) in the Lamm equation. Chaturvedi et al. [26] studied the concentration change of the NISTmAb (a medical protein) drug in the aqueous system by SEDFIT (an analysis software of non-ideal sedimentation) using the non-ideal sedimentation model; the study showed the distribution pattern of the macromolecular particles in time and space. In the field of geotechnical engineering, the soil particle size should not be made smaller with scale segmentation, and the non-ideal sedimentation model under high-speed centrifugal conditions is no longer suitable for simulating the original state of the engineering problem. By modifying the non-ideal sedimentation model, the constant-gravity sedimentation model was derived.

Therefore, based on the Lamm equation, this paper deduces a non-ideal sedimentation model that can simulate kaolinite under constant-gravity conditions and proposes a calculation method for the instantaneous settling rate (instantaneous flux of particle settling but not the movement rate of a specific particle) of a clay suspension at different depths in the measuring cylinder at any time. As kaolinite is a kind of natural fine-grained soil, it is difficult to flocculate and precipitate. Kaolinite has the property of antiflocculation—it can remain dispersed in water for a long time. Therefore, kaolinite is suitable for experimental observation purposes. In this paper, kaolinite is selected as the observation object to simulate the sedimentary characteristics of small-grained soil. In addition, soluble starch and apple pectin are chosen as flocculants for modification in this study. Kaya et al. [27] proposed that the higher the zeta potential, the faster the flocculation and sedimentation of kaolinite. Therefore, in this study, we use the zeta potential as a measure of the degree of flocculation. The effects of two kinds of polymers on the flocculation and sedimentation rate of kaolinite are analyzed by a sedimentation test. The effects of the addition of the flocculant on the decay coefficient, the instantaneous settling rate, and the sedimentation coefficient are studied here using the constant-gravity non-ideal sedimentation model.

## 2. The Constant-Gravity Non-Ideal Sedimentation Model

The non-ideal precipitation model was used to simulate the sedimentation characteristics of small particles under high-speed centrifugal conditions. Meanwhile, both vertical diffusion under gravity and radial deposition under a centrifugal force were considered, as shown in Figure 1a.

The change in concentration of small particles in the centrifugal tube with time and space satisfies the Lamm equation, as follows:(1)$$\frac{\partial c_{i}}{\partial t}=-\frac{1}{r}\frac{\partial }{\partial r}\left(s_{i}\omega^{2}r^{2}c_{i}-D_{i}\frac{\partial c_{i}}{\partial r}\right)$$

In Equation (1), *S_i_* denotes the sedimentation coefficient, *D_i_* represents the diffusion coefficient, *ω* is the angular velocity of the rotation of the centrifugal axis, *r* is the radial distance from any point in the centrifugal tube to the rotation axis, and *c_i_* is the concentration of particles in the water system and is a function of time (*t*) and radius (*r*). Based on the non-ideal precipitation model, SEDFIT software (National Institutes of Health, Bethesda, MD, USA) can be used to simulate the gradient relationship between *c_i_*, the radius (*r*), and the time (*t*) of the particle concentration in water.

In this study, kaolinite particles with a particle size *d* ≤ 75 μm were selected to simulate the precipitation process of fine-grained soil. Under macroscopic stress conditions, the kaolin particles in an aqueous solution can be thought to be only affected by gravity and buoyancy under high-concentration conditions. At a certain stage of the precipitation process, the concentration of kaolinite in the supernatant was relatively low, and the mutual exclusion force of negatively charged particles was balanced with gravity; at this stage, the influence of diffusion should be considered. This study mainly focuses on the stage when sedimentation occurs significantly, so only the vertical forces on the soil particles are considered, as shown in Figure 1b. We take *D_i_* = 0.

The constant-gravity non-ideal sedimentation model assumes that the kaolinite particle concentration is equal across the same height in the measuring cylinder, and $\omega^{2}r=a$, where *a* is the combined acceleration in the direction of gravity (gravity and buoyancy). The change in kaolin concentration satisfies the following equation:(2)$$\frac{dc_{i}}{dt}=\frac{2ac_{i}}{h}s\left(h,t\right)$$
(3)$$a=\frac{\left(\rho_{s}-\rho_{w}\right)Vg}{\rho_{s}V}=\frac{\left(\rho_{s}-\rho_{w}\right)}{\rho_{s}}g$$
where *c_i_* denotes the suspension concentration, *t* refers to time, *h* represents the vertical distance from the surface of the suspension to any point inside the suspension, *a* is the acceleration of kaolinite particles which can be calculated by Equation (3). $\rho_{s}$ is the density of kaolinite particles, and $\rho_{w}$ is the density of water. Here, the values of $\rho_{s}=$ 2.6 g/cm^3^ and $\rho_{w}\;$= 1 g/cm^3^ are taken. We integrate both sides of Equation (2) to get the following:(4)$$\int^{}s\left(h,t\right)dt=\int^{}\frac{h}{2a}\frac{1}{c_{i}}dc_{i}$$

In solving the partial differential equation of Equation (4), we can obtain the following:(5)$$S\left(h,t\right)=\frac{h}{2a}\ln c_{i}+C$$
In Equation (5), *S*(*h*,*t*) is the antiderivative of precipitation coefficient *s*(*h*,*t*). For the determination of the constant (*C*), the precipitation coefficient *s* = *v*/*g* is substituted into Equation (4), so as to obtain the following:(6)$$\int^{}\frac{v\left(h,t\right)}{a}=\int^{}\frac{h}{2a}\frac{1}{c_{i}}dc_{i}$$

To solve the partial differential equation of Equation (6), the following can be obtained:(7)$$l\left(h,t\right)=\frac{h}{2}\ln c_{i}+C_{1}$$
In Equation (7), *l*(*h*,*t*) is the antiderivative of *v*(*h*,*t*) and represents the displacement of the particle from the initial position (*h*) in time. According to the boundary condition *l*(*h*,0) = 0, the following can be obtained:(8)$$C_{1}=\frac{h}{2}\ln c_{0}$$
In Equation (8), *c*_0_ refers to the initial concentration of kaolinite in the suspension, and thus,
(9)$$C=\frac{h}{2a}\ln c_{0}$$

We substitute Equation (9) into Equation (4) to obtain the following:(10)$$S\left(h,t\right)=\frac{h}{2a}\ln\frac{c_{i}}{c_{0}}$$

Using the derivative Equation (10), the relation of sedimentation coefficient *s*(*h*,*t*) can be obtained:(11)$$s\left(h,t\right)=S_{t}^{\prime }\left(h,t\right)=\frac{h}{2ac_{i}}\frac{dc_{i}}{dt}$$

Equation (11) is the formula that is used to calculate the sedimentation coefficient (*s*) in the constant-gravity non-ideal sedimentation model, and it is a key index to measure the settling rate. The unit of the sedimentation coefficient is Svedbergs or seconds, and 1 Svedberg = $10^{-13}$ s. In this paper, the units of the sedimentation coefficient were, uniformly, seconds.

During the experiment, the *c_i_*(*h*,*t*) measurement satisfied the change rule of the sedimentation coefficient *s*(*h*,*t*).

In addition to the sedimentation coefficient (*s*), there was an important parameter in the model—the decay coefficient (*α*). It is not only a parameter reflecting the settling rate of suspension but also an index to evaluate the characteristics of the additional flocculant. By collating and fitting the experimental data, it can be seen that the change rule of the soil particle concentration in the suspension roughly satisfies the decay function, as follows:(12)$$c_{i}=c_{0}e^{\alpha t}$$

In Equation (12), *c*_0_ refers to the concentration of kaolinite in the suspension at *t* = 0. In this experiment, *c*_0_ = 6 g/L; *α* is the decay coefficient, and its value is negative. The value is related to the performance of the flocculant and the position of the measuring point.

After the measurement and data analysis, it was concluded that in the stage of rapid flocculation and the precipitation process, the value of the decay coefficient (*α*) was distributed in a logistic curve at different heights (*h*) in the measuring cylinder, and its variation law satisfied the following function:(13)$$\alpha =\frac{\alpha_{1}-\alpha_{2}}{1+{\left(\frac{h}{h_{0}}\right)}^{p}}+\alpha_{2}$$
In Equation (13), *α*_1_ represents the value at the surface of the measuring cylinder; *α*_2_ represents the value at the infinite of depth *h*; *p* is the curve shape index, which controls the growth rate of the decay coefficient (*α*) of the dependent variable; and *h*_0_ is the inflection point coordinate of the logistic function.

## 3. Materials and Methods

### 3.1. Observation Test of Small Particle Precipitation Characteristics

Malaysian kaolinite was selected for the experiment. The particle size distribution test showed that 99.95% of the soil particles had a particle size less than 0.075 mm. The particle size met the test requirements for simulating the precipitation characteristics of fine-grained soil. Furthermore, three different flocculants (soluble starch, apple pectin, and NaCl) were selected to compare the catalytic effect of flocculation.

Before the experiment, the Malaysian kaolinite needed to be processed with Na-homogenized treatment so that we can eliminate any interference from impurity ions in the later experiment. First, the Malaysian kaolinite was soaked in deionized water and stirred with NaCl solution. After 10 min of stirring with a Yunbang, 2500 blender (Jinhua, China), it was set aside to allow precipitation to occur. The next day, the supernatant was extracted with a syringe, and the conductivity of the supernatant was measured. We repeated this procedure until the measured conductivity was less than 10 μs/cm, and the supernatant was considered to be free of impurity ions. The kaolinite was dried and ground after being treated with sodium ion purification (Na-homogenized).

First, the observation experiment on the precipitation characteristics of modified kaolinite was carried out. Soluble starch and apple pectin (plant extracellular polymer) were selected as the added polymer flocculant, and NaCl was selected as the control group for observation.

The observation experiment was divided into three groups according to the types of flocculants, namely the NaCl group, soluble starch group, and apple pectin group. The soluble starch group and apple pectin group were divided into five groups according to the flocculant concentrations, namely 0.01, 0.05, 0.1, 0.5, and 1 g/L, respectively.

The specific steps are as follows:Weigh 8 g of Malaysian kaolinite, and make it into a 100 mL kaolinite suspension in the beaker.Use an S10 hand-held homogenizer (Shanghai Huxi Industrial Co., LTD, Shanghai, China) to shear the suspension at high speed. At a speed of 5000 r/s, the suspension continuously shears at high speed for 60 s to disperse the kaolinite into a single particle. To prevent the damage of the polymer structure, a polymer flocculant is added after the shear dispersion process.Slowly pour the prepared suspension into the 100 mL measuring cylinder, and keep the scale at 100 mL using a glue head dropper.Start timing after shaking, and observe and record the thickness of the supernatant and sediment. Data (suspension level scale and sediment thickness) were recorded every 12 h for 35 days.

Imai et al. [28] studied the sedimentation mechanism and sediment formation of dilute clay–water mixtures and proposed that the sedimentation process can be divided into three stages—the flocculation stage, sedimentation stage, and consolidation stage. In the experiment we found that the three stages occur almost simultaneously at the beginning of the deposition of particles, as shown in Figure 2. At the initial flocculation stage (*t* = 0), the suspended single particles began to flocculate, form aggregates, and settle rapidly. When the flocculation sedimentation was completed (*t* = *t*_0_), the sediments in the bed continued to consolidate. The sediments were generally large kaolinite aggregates.

Over time, the suspension in the measuring cylinder gradually changed from top to bottom into three layers—the supernatant, the suspension, and the bed layer. With the advance of the deposition process, the intermediate suspension layer gradually disappeared, leaving only the supernatant layer and the bed layer. When the precipitation process reached *t* = 50,000 min, the observation experiment was terminated.

The zeta potential was measured with a Shanghai Zhongchen JS94Hzeta potentiometer (Shanghai, China), and the value of the zeta potential was calculated based on the principle of microelectrophoresis pulse balance. The steps are as follows:Take about 2 mL of the diluted suspension sample, insert the electrode and pass it through the alternating current. Under a microscope (Shanghai, China), observe that the soil particles in the suspension will migrate back and forth in the horizontal direction.When the migration process tends to be stable (the particles move back and forth in the same region), take a screenshot and read the corresponding horizontal and vertical coordinates of the soil particles in the two screenshots before and after, and calculate the zeta potential on the surface of the soil particles at this time.

### 3.2. Settlement Rate Test Experiment

The settlement rate test groups are given in Table 1.

Soluble starch and apple pectin were used as the added flocculant in the experimental group, and NaCl was used as the added flocculant in the control group. In the experimental group, soluble starch that was dissolved in water, resulting in gelatinization, and so it had a certain agglomeration capacity. Apple pectin is rich in various amino acids. After the amino was dissolved in water, weak ionization occurred, and a large number of anions were generated, further reducing the zeta potential of the system and dispersing the soil particles in the solution. After the NaCl control was dissolved in water, the ionized cation neutralized the negative charge on the surface of the soil particles in large quantities, reduced the mutual exclusion between particles, and accelerated the flocculation and sedimentation process of the soil particles rapidly.

Based on the principle of light reflection, the REMOND R801 (Wuhan, China) sludge concentration meter was used to determine the concentration of the soil particles. Because of the laboratory light intensity and the material transmittance, it was necessary to calibrate the sludge concentration meter. The specific steps were as follows:The prepared suspensions of each experimental group were cut with a homogenizer at high speed so that the kaolinite particles can be distributed in a single particle and in uniform contact with the flocculant, which were then evenly poured into a measuring cylinder with a capacity of 5000 mL.During the measurement of the soil particle concentration, the sludge concentration meter was used to measure the concentration at each 500 mL scale, and the sensor was stationary at the corresponding scale for about 2 min before each reading. The data were recorded after the sensor reading had been stabilized.

Figure 3 shows the experimental effect. During the experiment, the concentration was measured at 09:00 and 21:00 each day, and the illumination intensity around the concentration sensor was kept uniform as far as possible to reduce the deviation caused by the difference in illumination intensity.

## 4. Results

Figure 4 shows the sedimentation process of kaolinite under the action of different concentrations and different kinds of flocculants (soluble starch and apple pectin; h–t curve). Figure 5 shows the zeta potential change of two groups.

Through experiments, the following were found:In the sedimentation curves shown in Figure 4, the deposition process is reflected by both the supernatant calibration curve and the bed calibration curve. At the beginning of the flocculation settling, the scale of supernatant was 100 and that of the bed was 0. With the continuous process of flocculation settling, the scale of the supernatant decreased gradually, and that of the bed rose slowly. When the deposition process was completed, the two scales tended toward the same value.It can be seen from the sedimentation curves in Figure 4a,b that in the experimental group with soluble starch as the flocculant, the flocculation and precipitation process of kaolinite significantly intensified with the increase of soluble starch. In the experimental group with apple pectin as the added flocculant, the flocculation and precipitation of kaolinite slowed down with the increase in apple pectin.It can be seen from Figure 5 that the settling rate of the kaolinite particles was negatively correlated with the zeta potential, and the types of flocculants added affected the change of zeta potential with the concentration. The zeta potential of the soluble starch group increased with the addition of the flocculant. The zeta potential value in the experimental group of apple pectin was inversely proportional to the amount of added flocculant. The zeta potential of the NaCl control group increased significantly with the increase of NaCl.

To further understand the variation law of the kaolinite particle sedimentation rate, this study adopted the constant-gravity non-ideal sedimentation model and studied kaolinite particle concentration (*c*) as the physical quantity that reflects the sedimentation rate.

The decay coefficient (α) and the sedimentation coefficient (*s*) have been proposed in this paper. The decay coefficient (α) is the parameter that reflects the flocculation performance of the additive, and the sedimentation coefficient (s) is the parameter that reflects the settling rate of the soil particles.

The study simulated the change rule of the kaolinite concentration after adding the flocculant, as shown in Figure 6. By calculation, the R-square of simulation is shown in Table 2. The curve was calculated from Equation (12), and the overall trend was an exponential decay. It can be seen from the figure that, for each applied flocculant, the greater the depth (*h*), the slower the decay rate of concentration (*c*). Moreover, the settling rate of the kaolinite particles varied with the different types of flocculants.

As shown in Table 2, the coefficient of determination of the concentration curves’ simulation is close to 1. The results prove that the simulation has a high accuracy. The decay rate of the concentration (*c*) can be measured by the decay coefficient (*α*). This paper has measured and analyzed the variation of the decay coefficient (*α*) with different depths, and its law conforms to the logistic curve, showing a rapid increase in the early stage and stabilization in the later stage, as shown in Figure 7. The curves in Figure 7 are fitted by Equation (13).

In order to more intuitively reflect the soil particle concentration (*c*) change with time (*t*), we used the established model (Equation (12)) to simulate the soluble starch group sedimentary concentration curves, as shown in Figure 8, to represent the changes of concentration with time and depth. Compared with the measurement results, the modelling results have some deviations. The deviations are concentrated in the second half of the curves because the empirical formula (Equation (12)) is not accurate enough.

Furthermore, a method for calculating the sedimentation coefficient (*s*) and the instantaneous sedimentation rate of small particles at a specific height in the water was proposed. Through the constant-gravity non-ideal precipitation model (Equation (11)), the variation of the sedimentation coefficient (*s*) with time (*t*) and depth (*h*) can be plotted. As can be seen from Figure 9, as the depth increases, the sedimentation coefficient (*s*) increases synchronously. The sedimentation coefficient (*s*) at a specific height decreases gradually with the passing of time. When *s* approaches 0, the flocculation and deposition process is considered completed.

## 5. Case Analysis

To prove the accuracy of the model, a case involving a carbon nanotube suspension precipitation experiment [29] was cited for verification, where Hu et al. measured the concentration of a carbon nanotube suspension with flocculant. The relationship between the concentration of carbon nanotubes and time was obtained, as shown in Figure 10.

The model was used to calculate the data in this case, using Equation (11), and the change in the sedimentation coefficient of the suspensions as a function of time was calculated under for different flocculants, as shown in Figure 11.

From Figure 11, we can observe that the sedimentation coefficient of each group gradually decreased with time, however at very different rates. The sedimentation coefficient *s* of groups CNTs-SDS and CNTs-SDBS decreased to 0 when *t* approached 400 h, and that of groups CNTs-Brij35, CNTs-CTAB, and CNTs-SLS decreased to 0 when *t* approached 150 h.

Comparing Figure 10 and Figure 11, we can see that the time at which the concentration of suspension becomes stable is consistent with the time the sedimentation coefficient approaches 0.

## 6. Discussion

According to Figure 4 and Figure 5, different types of flocculants have different catalytic effects on the flocculation–agglomeration of kaolinite particles. Soluble starch dissolves in water and gelatinizes, and the hydrogen bond between the starch molecules breaks and partially neutralizes the negative charge on the surface of the soil particles, making the zeta potential rise and kaolinite particles flocculate and aggregate. Apple pectin is an extracellular polymer secreted by plant cells, and it is an important component of the plant cell wall. The apple pectin molecule contains a large number of amino functional groups, and after dissolving in water, a reversible weakly ionized reaction happens, where $R-NH_{2}+H_{2}O\overset{\;}{\Leftrightarrow }R-NH_{3}^{+}+OH^{-}$. The ionization of hydroxyl ions reduces the zeta potential of the solution and results in slow flocculation sedimentation. Therefore, with the increase in the concentration of the apple pectin, the process of flocculation and the precipitation of kaolinite particles was delayed and the sedimentation rate slowed.

The constant-gravity non-ideal sedimentation model proposed in this study can be used to quantitatively evaluate the effect of polymer addition on the sedimentation flocculation of kaolinite particles. According to Figure 6, the change rate of concentration (*c*) will be significantly different according to the types and concentrations of flocculants, and the decay coefficient (*α*) can be used to describe the change in particle concentration. The decay coefficient (*α*) is a parameter that characterizes the performance of the flocculant. Generally, the larger the absolute value of (*α*), the better the flocculant behaves. Because the flocculation mechanisms of different flocculants differ (such as soluble starch through gelatinization and apple pectin through ion exchange after weak ionization of amino functional groups), their decay coefficients will be quite different. The value is related to the type of flocculant, the amount of flocculant added, and the section depth of the measuring cylinder.

According to the measured concentration *c*, the *s–t–h* curves can be drawn, as shown in Figure 9. In the normal gravity precipitation model, the sedimentation coefficient *s* is an important parameter to measure the instantaneous velocity of the soil particles at a specific location. The larger the sedimentation coefficient *s*, the stronger the agglomeration effect of flocculant on the single kaolinite particles. Figure 10 shows the *s*–*t*–*h* curves of the apple pectin group at 0.1 g/L. At any height specified in the measuring cylinder, the sedimentation coefficient *s* tends to decrease gradually over time. When *s* approaches 0, the precipitation process is considered to be completed.

In Section 5, a previous example of carbon nanotube suspension deposition was selected. By comparing the calculated results with the conclusions for that case, it is proven that the model has a high accuracy.

## 7. Conclusions

In this paper, a constitutive model was proposed to simulate the non-ideal settlement of the clay particles under constant-gravity conditions. The key parameters (decay coefficient *α* and the sedimentation coefficient *s*) of the instantaneous settling rate of the reaction were derived from the constant-gravity non-ideal sedimentation model. The flocculation of kaolinite by different kinds of polymers was obtained through experimental evaluation of the deposition characteristics of kaolinite. The following conclusions can be drawn from the experimental results.

Compared with the calculation method of the settling rate based on Stokes’ law, the model proposed in this paper avoids a series of difficulties, such as uneven suspension and the difficult observation of the boundary line between supernatant and suspension, and it can more accurately and intuitively obtain the variation law of concentration at different depths.Because of the characteristics of the added flocculant, different flocculants have different flocculation effects on the kaolinite particles. For example, after soluble starch is dissolved in water, the intermolecular hydrogen bond breaks and gelatinization occurs. This reaction promotes the flocculation of kaolinite particles, to some extent. After the apple pectin was dissolved in water, the amino functional groups in the macromolecules weakly ionized, and the generated hydroxide ions further reduced the zeta potential in the aqueous system and slowed the flocculation process as well as the precipitation of soil particles.The zeta potential reflects the electrification of the surface of small particles, and the more neutral the charge, the more likely it is to flocculate and settle. However, because the different flocculants have different flocculation mechanisms—for example, the gelatinization of soluble starch dissolved in water will not significantly change the electrification of the surface of small particles—the zeta potential has limitations regarding its control of the flocculant flocculation capacity.The accuracy of the model was verified. As shown in Table 2, the value of the R-square ranges from 0.93296 to 0.96644, which has a high goodness of fit. This model can be used to calculate the case of carbon nanotubes suspension sedimentation, and the results are in good agreement with the conclusions of that study. It was proven that the model has a high application value in the study of sedimentary characteristics under a constant-gravity environment.Two important parameters were proposed in this paper—sedimentation coefficient *s* and decay coefficient *α*. Sedimentation coefficient *s* reflects the settling rate of kaolinite at any height in the measuring cylinder, and its value depends on the position in the measuring cylinder and the effect of the flocculant on the aggregate flocculation of kaolinite particles. It was found that sedimentation coefficient *s* increases with the increase in depth in the measuring cylinder. Decay coefficient *α* is a parameter that characterizes the material properties of the flocculant, and the coefficient depends on the physicochemical properties of the flocculant itself. Furthermore, the model can obtain the performance parameters of various additional flocculants and is helpful for the further study of the precipitation characteristics of small particles.

## Figures and Tables

**Figure 1 materials-13-03785-f001:**
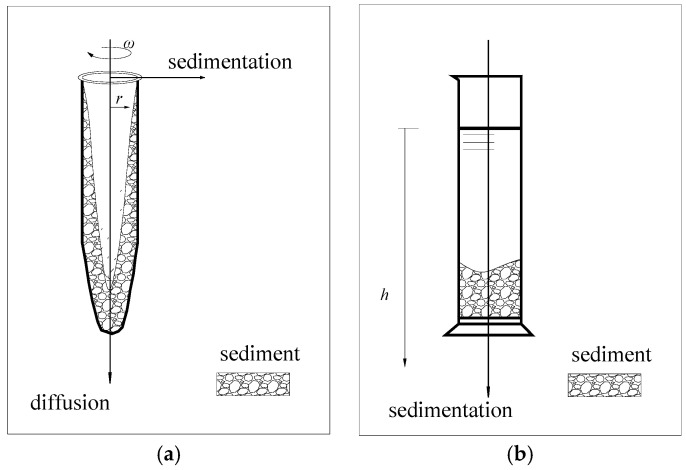
(**a**) The non-ideal sedimentation model and (**b**) the modified constant-gravity sedimentation model.

**Figure 2 materials-13-03785-f002:**
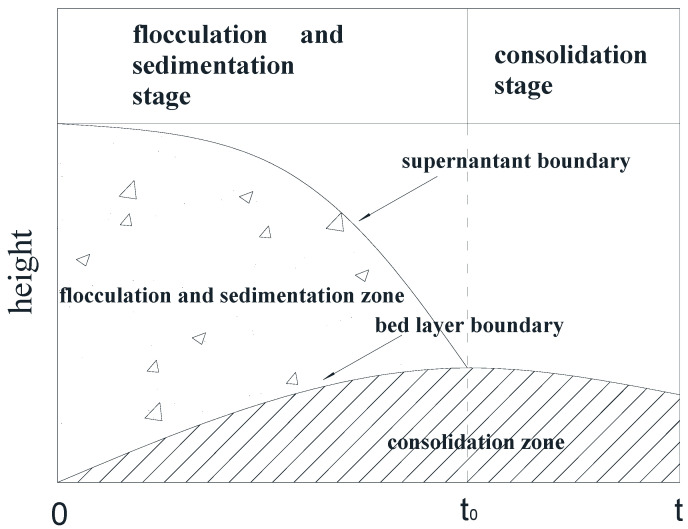
Flocculation and sedimentation of kaolinite particles in the measuring cylinder.

**Figure 3 materials-13-03785-f003:**
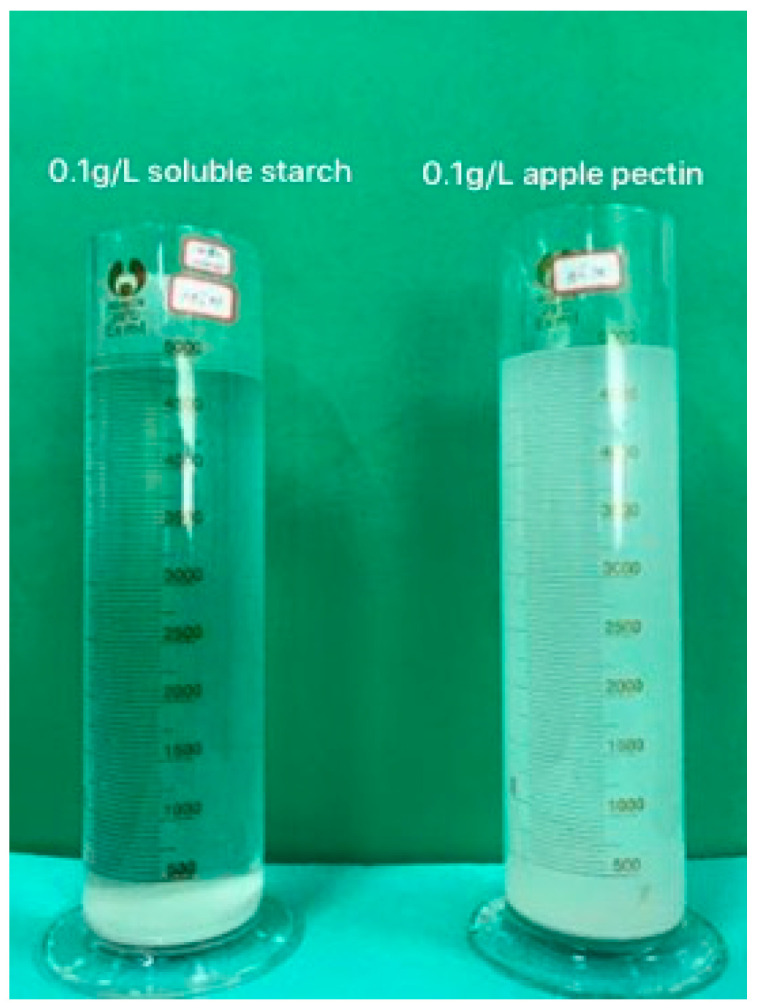
Experimental photo.

**Figure 4 materials-13-03785-f004:**
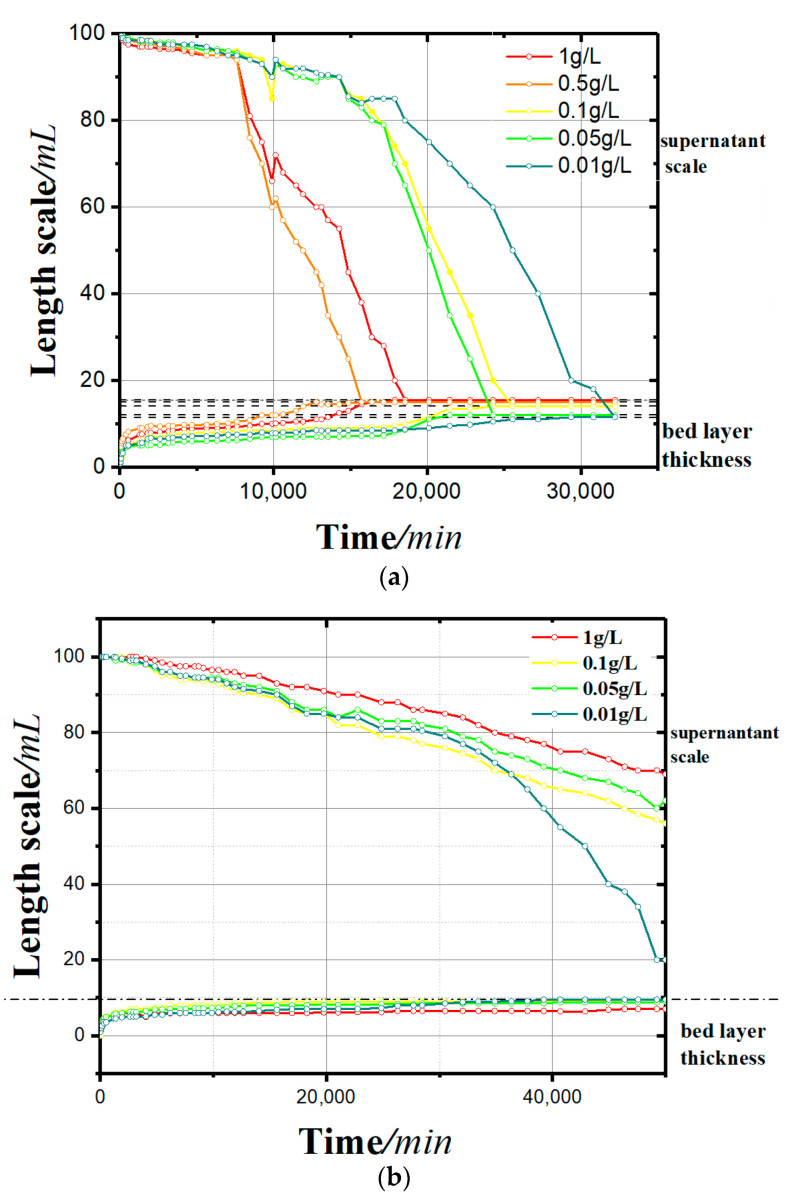
Sedimentation curves: (**a**) with a soluble starch flocculant and (**b**) with an apple pectin flocculant.

**Figure 5 materials-13-03785-f005:**
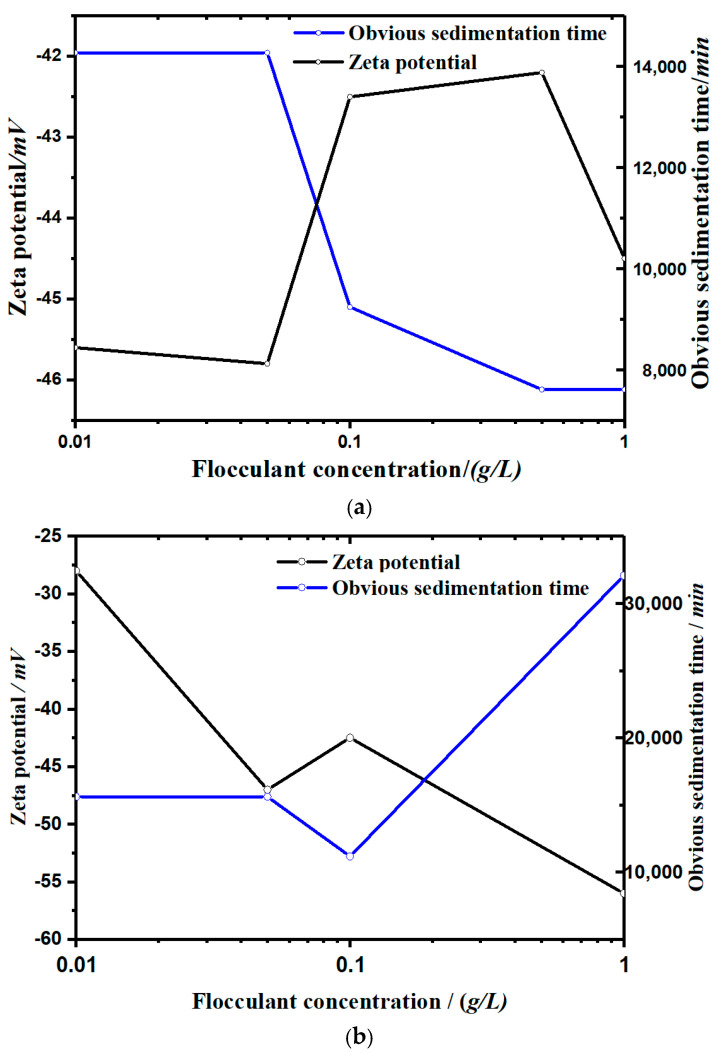
The zeta potential corresponding to the obvious sedimentation time in each concentration group: (**a**) with a soluble starch flocculant and (**b**) with an apple pectin flocculant.

**Figure 6 materials-13-03785-f006:**
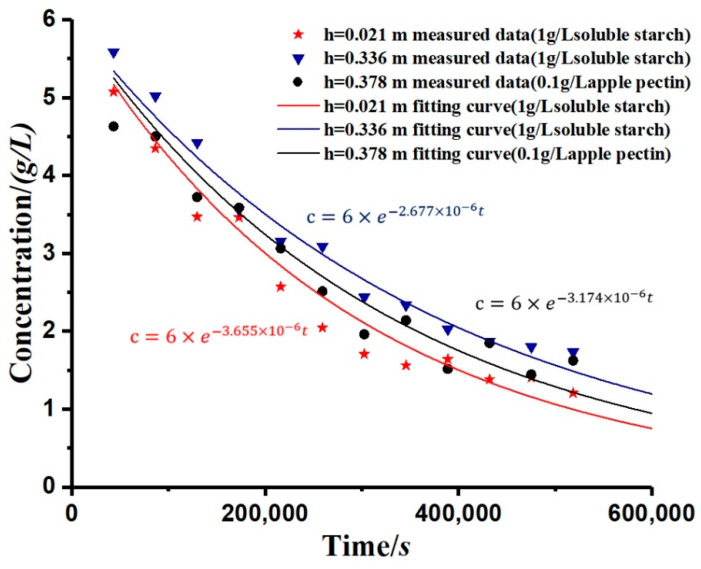
The measured data and modelling curves in each concentration group.

**Figure 7 materials-13-03785-f007:**
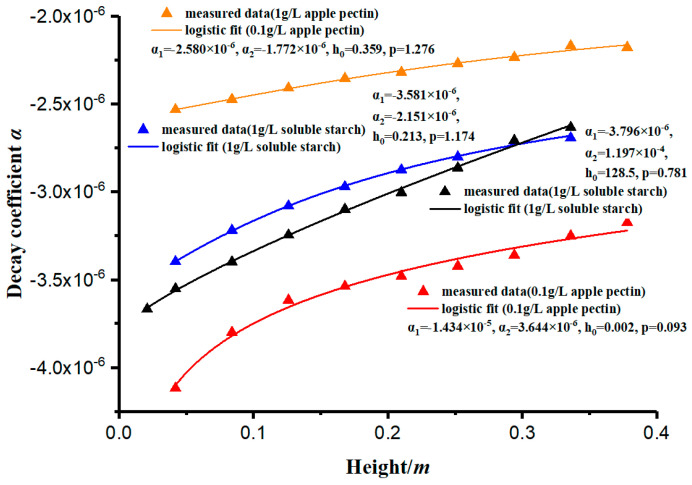
The logistic fitting of the decay constant (*α*).

**Figure 8 materials-13-03785-f008:**
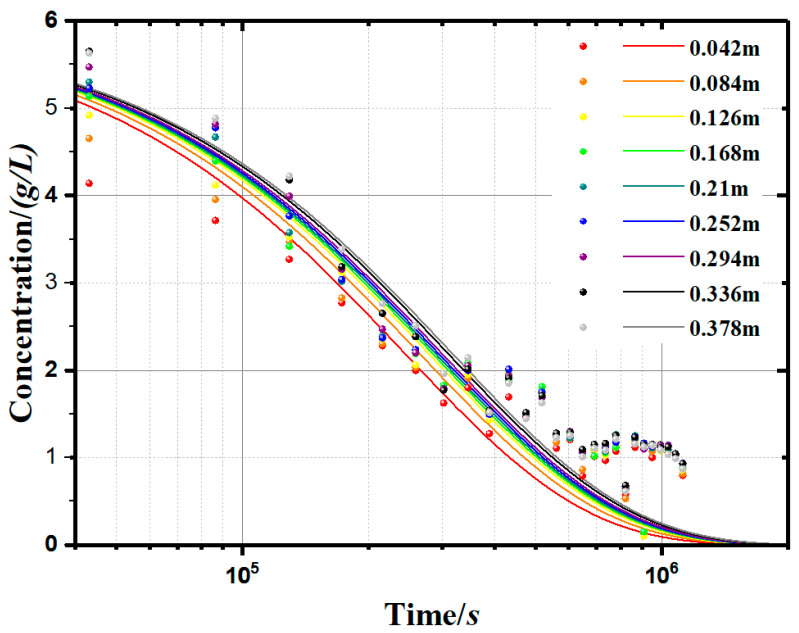
Measurement and modelling results of kaolinite suspension concentration (0.1g/L apple pectin experimental group).

**Figure 9 materials-13-03785-f009:**
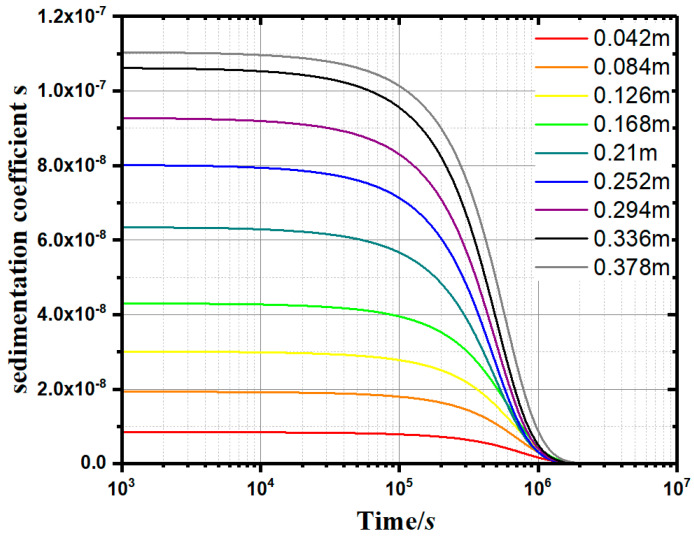
Sedimentation coefficient curves (0.1g/L apple pectin experimental group).

**Figure 10 materials-13-03785-f010:**
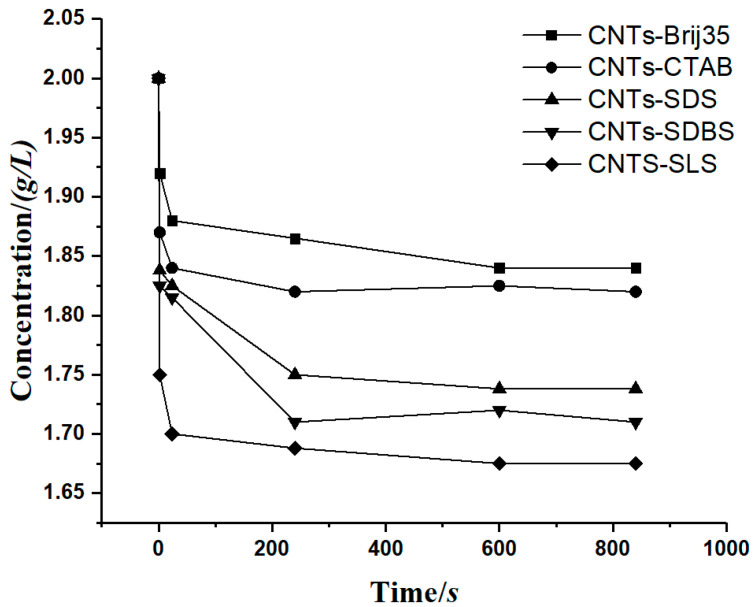
The relationship between the carbon nanotube suspension concentration and time.

**Figure 11 materials-13-03785-f011:**
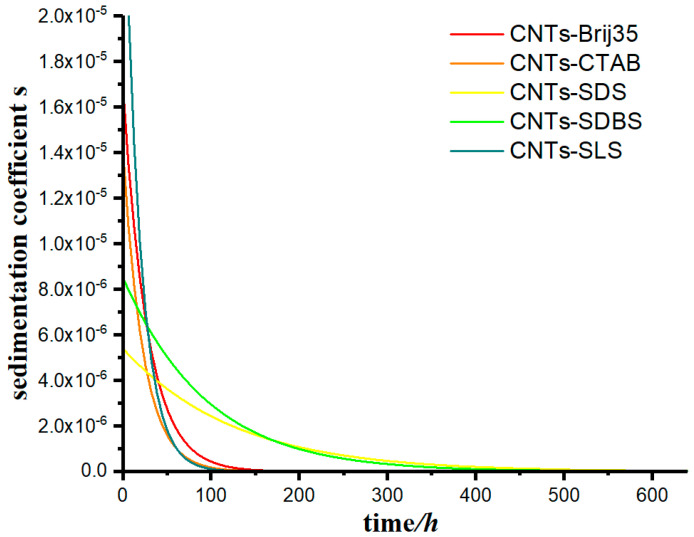
The change in the sedimentation coefficient of carbon nanotube suspension as a function of time for various flocculants.

**Table 1 materials-13-03785-t001:** Experiment groups for testing the sedimentation rate.

Experimental Group	Kaolinite/g	Deionized Water/L	Flocculant
1	30	5	0.1 g/L soluble starch
2	30	5	1 g/L soluble starch
3	30	5	0.1 g/L apple pectin
4	30	5	1 g/L apple pectin
5	30	5	0.1 mol/L NaCl

**Table 2 materials-13-03785-t002:** The coefficient of determination of the concentration curves’ simulation.

Height/Flocculant	0.021 m/Soluble Starch (1 g/L)	0.336 m/Soluble Starch (1 g/L)	0.378 m/Apple Pectin (0.1 g/L)
R^2^	0.966	0.979	0.933

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
