# Peer review of "Modelling and Experimental Investigation on the Settling Rate of Kaolinite Particles in Non-Ideal Sedimentation Stage under Constant Gravity"

_materials, 2020, doi:10.3390/ma13173785_

Round 1
Reviewer 1 Report
Because of the significance in engineering practice, the sedimentation behavior of flocculated colloid suspension can be regarded as an important subject. So far many researches have been reported. This research is the extension of authors earlier work ref.17, with an indication of new experimental results of new polymer flocculant. Although the references cited in Ref.17 is appropriate, the present article has skipped several important one. They are:
Imai G., Experimental studies on sedimentation mechanism and sediment formation of clay materials, Soils and Foundations, 21(1981) 7-20.
Imai G., Settling behavior of clay suspension, Soils and Foundations, 20 (1980) 61-77.
Michaels A.S., Bolger J.C., Settling rates and sediment volumes of flocculated kaolin suspensions, Industrial & Engineering Chemistry Fundamentals, 1 (1962) 24-33.
Author may also find new idea from the following review and cited articles.
KONA Powder and Particle Journal /Doi:10.14356/kona.2020019
Due to these imperfectness, I will not recommend the publication.
My simple question for the definition of Author is the terminology of
non-ideal sedimentation. What ideal sedimentation means.
Reviewer 2 Report
General comments
The authors investigate settling process of kaolinite particles in non-ideal sedimentation stage under constant gravity. I feel that the topic and the results are fit in ACS OMEGA. However, there are several points that have to be clarified and improved before it can be accepted.
Specific comments
Line 92. Why the authors choose soluble starch and apple pectin as flocculants in this experiment?
Line 203. How the authors measured the sedimentation process into 3 layer? DO you have any standard concentration or method to divide sedimentation process. And How is the accuracy? In Figure 4, the difference of bed layer thickness looks less than 1 mm.
Line 264. The unit Y axis of Figure 4 should be change into length scale.
Line 279. The authors mention that settling rate of the kaolinite particles was negatively correlated with the zeta potential. However, the experimental results shown in Figure 5 do not show clear negative correlation. Especially Figure 5(a) seems positive correlation.
Line 300. Units of h should be added in the Legend of Figure 6.
Line 300. In Figure 6, only 2 cases of experiment was simulated. In order to show a reproducibility of the model, the authors should include all modelling results in the Figure.
Line 310. The authors should include all modelling results in Figure 7.
Line 312-346. The authors shows Figures 10 and 11 to prove accuracy of the model. However, the particle and flocculants used in the experiment is different. In my opinion, addition of measured data in Figure 8 is the best way to prove accuracy of the model.
Line 312. For the benefit of the reader, equation number should be given.
Line 316. For the benefit of the reader, equation number should be given.
Round 2
Reviewer 2 Report
General comments
The authors investigate settling process of kaolinite particles in non-ideal sedimentation stage under constant gravity. I feel that the topic and the results are fit in ACS OMEGA.
The revised Figure 8 shows show low accuracy of the model. However, authors did describe the reason of low accuracy or limitation of the model.
And data used in the figure 5 was reversed during the review process.
Overall, I would not suggest the publication of this paper on the Materials.
Author Response
please see the attachment.

This manuscript is a resubmission of an earlier submission. The following is a list of the peer review reports and author responses from that submission.
Round 1
Reviewer 1 Report
The purpose of your paper is to present a model to predict sedimentation of kaolin. The new model uses a more complicated approach than previous model and accounts for non-uniform distribution of particles with depth as well as a less well-defined transition across supernatant (incorrectly spelled in manuscript) and bed boundaries. However, you are missing some elements that would make more understandable and direct paper.
- State the objective of your paper and what you plan to show the reader at the beginning. I did not know where you were going and what you intended to show other than a new model, kaolin and two flocculants.
- If this is a new (hopefully better) model then show its predictions compared to older (hopefully less accurate) models.
- If you are fitting the model to your data, you absolutely must give a statistical value that shows goodness of fit. You may also want to compare this to previous models as well to demonstrate the superiority of your model.
- Your 3D graphs are poor. The only purpose of a 3D graph is to show something that a 2D graph cannot. Both of the 3D graphs are shown from the wrong perspective and do not demonstrate the 3D behavior of what you wish to show. This will take some time and patience to get right.
- Include some quantification of how much better your model is versus previous ones and include that in your results and conclusions.
- Can you predict the data from figure 5? This is the whole point of the article is it not?
- Can your model help predict zeta potential vs. flocculant concentration/sedimentation time ?
- Fig 6 is not a very convincing curve fit for your data. Why do you insist on the particular form of the curve? Do previous models do worse?
Please demonstrate how well your model works compared to others. If it does not then you can say the preliminary experimental work is promising and there will be additional testing and adjustments to the model.
Some comments are on the manuscript below

Reviewer 2 Report
In this article, the authors report on a model to simulate the non-ideal sedimentation of clay particles in an aqueous system under constant gravity. According to the authors, this model not only considers the inhomogeneity of the solute but also simulates the change in clay concentration with time during sedimentation. The sedimentation characteristics of kaolin clay in aqueous solutions of NaCl, soluble starch and apple pectin are studied experimentally. The results indicate that the model is capable of predicting the time required for complete sedimentation of the particles.
The beginning of this work involves a discussion on the concept of sedimentation of clay particles, including the importance of understanding involved non-ideality effects at high concentrations. This is definitely an important question and has high relevance in the field as illustrated by the authors. References are given on the sedimentation of clay particles and non-ideality in general. However, it is unclear to me how the authors’ approach performs against existing models taking into account non-ideality, such as the Gralen coefficient or the hindrance function. The latter is also used for highly dense dispersions. From my perspective, it would be necessary to discuss possible connections in the manuscript. Noteworthy, the aforementioned models allow prediction of the concentrations at any point in the cell / cylinder and not just the progress of the sedimentation boundary. Moreover, I must say that the theoretical considerations in the paper are somewhat odd as the non-ideality effects are not considered as an intrinsic dependence of the sedimentation coefficient on concentration but are instead fitted using an arbitrary function based on an empiric approach. At least some relation should be given to model-based approaches, which already exist in literature. Why was an empirical model chosen for the given purpose?
In the authors’ derivations, I further encountered some errors. E.g., the sedimentation coefficient lacks an appropriate unit. Equation (3) makes no sense as the units don’t fit. The unit of the kaolinite particle density is wrong.
Besides the scholarly presentation, spelling, grammar and punctuation should be carefully revised as they are flawed in many parts.
For the given reasons, I recommend a major revision. Once, all errors are corrected and the results are set in relation to existing models available in literature, I am happy to recommend the paper for publication.
Round 2
Reviewer 1 Report
I am still puzzled why the authors did not include the actual parameters to the curve-fit equations, such as figure 6. I reproduce it below with your data and my curve-fits with their equations and R^2 values. This is all I wanted for this figure. Note also that one of the curves does has a value of about exp(-2.6e05) not exp(-3.0e05) as stated in the paper.
See the attached file for Figure6,7,8
Figure 7 needs to have the Y-axis changed to show more decimals of number since all of the numbers show 3x10-06 in the figure. What is the equation for the curve in figure 7? You should have it in the figure along with R^2 as well.
Figure 8 and 9 are both unreadable, I would suggest that you just plot a 2-d family of curves instead as shown in the attached Figure8

Reviewer 2 Report
I would like to thank the authors for their response to my questions. Unfortunately, I'm still not convinced that the manuscript is suitable for publication. In particular, the criticism given in Point 1 was not addressed by the authors. While just the information provided in the manuscript was repeated in their answer, no comprehensive scientific discussion is provided by the authors. This, however, is crucial from my point of view to be at least able to estimate how well their new model performs against other existing approaches.
Eq. 3 is still not correct as rho - 1 does not make any sense when considering the unit of density. I believe that the authors would like to refer to the density of water which is approximatly 1 g/cm3. However, this is not clear and should thus be revised.
